# Genome Complexity Browser: Visualization and quantification of genome variability

**Alexander Manolov**[ID][☯]*, **Dmitry Konanov**[ID][☯], **Dmitry Fedorov**[ID], **Ivan Osmolovsky**[ID], **Rinat Vereshchagin, Elena Ilina**[ID]

Federal Research and Clinical Center of Physical and Chemical Medicine, Federal Medical and Biological Agency of Russia, Moscow, Russian Federation

☯ These authors contributed equally to this work.

* manolov@rcpcm.org

**Data Availability Statement:** All relevant data are within the manuscript and its Supporting Information files.

**Funding:** The authors received no specific funding for this work.

## Abstract

Comparative genomics studies may be used to acquire new knowledge regarding genome architecture, which defines the rules for combining sets of genes in the genome of living organisms. Hundreds of thousands of prokaryotic genomes have been sequenced and assembled. However, computational tools capable of simultaneously comparing large numbers of genomes are lacking. We developed the Genome Complexity Browser, a tool that allows the visualization of gene contexts, in a graph-based format, and the quantification of variability for different segments of a genome. The graph-based visualization allows the inspection of changes in gene contents and neighborhoods across hundreds of genomes, simultaneously, which may facilitate the identification of conserved and variable segments of operons or the estimation of the overall variability associated with a particular genome locus. We introduced a measure called complexity, to quantify genome variability. Intraspecies and interspecies comparisons revealed that regions with high complexity values tended to be located in areas that are conserved across different strains and species.

## Author summary

The comparison of genomes among different bacteria and archaea species has revealed that many species frequently exchange genes. Occasionally, such horizontal gene transfer events result in the acquisition of pathogenic properties or antibiotic resistance in the recipient organism. Previously, the probabilities of gene insertions were found to vary, with unequal distributions along a chromosome. At some loci, referred to as hotspots, changes occur with much higher frequencies compared with other regions of the chromosome. We developed a computational method and a software tool, called Genome Complexity Browser, that allows the identification of genome variability hotspots and the visualization of changes. We compared the localization of various hotspots and revealed that some demonstrate conserved localizations, even across species, whereas others are transient. Our tool allows users to visually inspect the patterns of gene changes in graph-based format, which presents the visualization in a format that is both compact and informative.

**Competing interests:** The authors have declared that no competing interests exist.

## Introduction

The genome is not simply the storage of gene sequences. Many sequence motifs and sequence patterns must be located at proper loci to ensure proper interactions between the molecular machines involved in basic cell processes (transcription, genome replication, and cell division) and the chromosome [1–3]. The interactions between these cellular processes and DNA molecules and the interplay among them govern optimal gene localizations and orientations [1]. New knowledge regarding the genome architecture, which represents the set of imposed constraints and favorable configurations for genomic objects [1], may deepen our knowledge of the basic cellular processes and facilitate the development of artificial genomes in the field of synthetic biology.

The non-random localization of different genes may be important, due to several factors. Genes located near the replication origin may have higher copy numbers in fast-dividing cells, which is known as the replication-associated gene-dosage effect [4, 5]. Chromosomal folding can bring genes located in different regions of the chromosome close to each other in 3D space, which can be beneficial for genes that encode a regulator and its target genes [6, 7]. The gene expression effects of global regulators, such as the histone-like nucleoid-structuring protein (H-NS), have been shown to depend on the location of the target genes [8], and the transcriptional propensity also varies, depending on the position of the gene within the chromosome [9]. The cooperative effects of RNA polymerases [10] and supercoiling propagation may play roles in the transcriptional regulation of neighboring genes.

Horizontal gene transfer (HGT) events are preferentially localized in hotspots, which are chromosomal loci in which changes are observed much more frequently than in other regions [11–13]. Although disruptions in genome architecture may result in the decreased fitness of an organism, changes can be introduced in some regions of the chromosome without inducing negative effects. To our knowledge, no currently available computational tool is capable of performing quantitative estimations of variability along the chromosome. Such a tool is necessary to deepen our knowledge of the factors determining genome variability and stability.

Here, we present the Genome Complexity Browser (GCB), a tool that allows the estimation of local genome variability and visualization of gene rearrangements. Both tasks are performed using graph-based representations of gene neighborhoods within a set of genomes. Local genome variability is evaluated using the metric introduced here, which we referred to as complexity. Complexity profiles may be used to identify hotspots of horizontal gene transfer or other local gene rearrangement events. The graph-based visualization available in GCB allows the analysis of patterns among genome changes events and the detection of persistent or variable gene combinations (e.g., variable and conservative regions of operons).

## Materials and methods

### Graph construction

The input for this step is the set of genomes, with inferred orthogroups. The algorithm for graph construction is the following: each orthogroup is represented as a node, and two nodes are connected by a directed edge if the corresponding genes are located sequentially in at least one genome in a set. The weight of the edge is calculated as the number of genomes in which corresponding genes are adjacent (see Fig 1A and 1B). Graph objects and their methods are implemented in the gene-graph-lib library for Python 3, and more information can be found within the library documentation at https://github.com/DNKonanov/gene_graph_lib.

Because GCB uses directed graph-based representations of gene order, all genomes in a set are first coaligned, to achieve the same orientation throughout the set. This step is performed

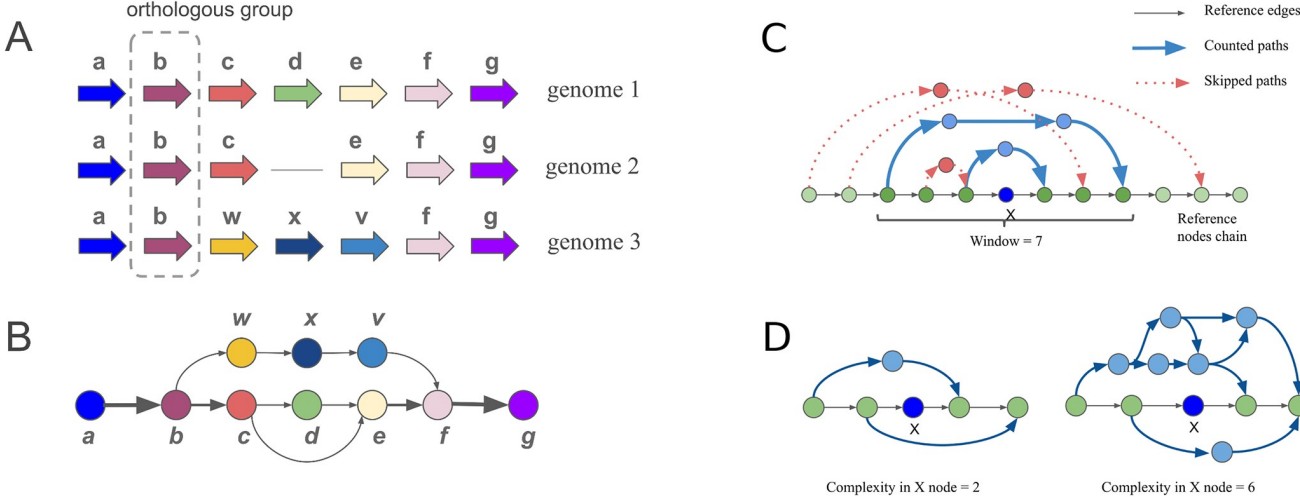

**Fig 1. Principal scheme used for the graph-based representation of gene order in a set of genomes and the genome variability estimation approach.** To construct a graph, each orthogroup is represented as a node. Nodes are connected by a directed edge if the corresponding genes are arranged sequentially in at least one genome in the set. A. Genomes 1, 2, and 3 represent three different hypothetical genomes. The arrows represent genes, and genes within the same orthogroup have the same color and letter designation. B. Graph-based representation of the three genomes shown in A). The weight of the edge (arrow width) is calculated as the number of genomes in which corresponding genes are located sequentially. C. Deviating paths for node X are defined as paths in the graph which bypass the node X and are connected with the section of the reference node chain limited by the window parameter. D. Two examples of counting deviant paths are shown. X is the considered node, deviating paths are shown with blue lines. Complexity value is defined by the number of deviating paths.

automatically by default, but may be optionally skipped because it takes a lot of time when a large number of draft genomes are included in the analysis (the runtimes are shown in the results section). The algorithm used for this step can be found in S1 Listing.

Orthogroups may include paralog genes [14], and some modifications to the basic process used for graph construction are necessary in this case. Two methods are suggested in GCB. The first method does not include the orthogroup in the graph for genomes containing paralogs (however, the same orthogroup will be included in the graph for genomes containing no paralogs), shown in S1A Fig. Thus, if some orthogroup contains two paralogs from only one genome, then the orthogroup will be excluded from graph for this genome, while still being included for all other genomes. This approach has the advantages of simplicity and clear output, although some genes will be missed in the graph. The second approach is to "orthologize" paralog genes: for each set of paralogous genes with a unique context, a graph node with a unique suffix will be created (S1B Fig). GCB uses the first approach by default, whereas the second approach is available as an option, in both the command line and browser-based versions.

## Genome complexity definition

We introduce a measure called complexity for the quantification of local genome variability. Complexity values are calculated against one reference genome in the set. This genome is extracted from the graph as a simple chain of nodes, called the reference chain (Fig 1C). To calculate complexity in node X, nodes from the reference chain, in the range ±*window*, around node X are selected, and the complexity is defined as the number of distinct paths in the graph that do not contain the node X but that start and finish in the nodes from the selected range (deviating paths), as shown in Fig 1C.

Complexity computing is an iterative algorithm that generates a set of possible deviating paths from each node in the reference genome (Algorithm1) and when a new unique deviating

path is found, the algorithm adds 1/*window* to the complexity values of all nodes in the reference between the nodes that represent the start and end of the deviating path (Algorithm2).

**Algorithm 1**: FIND PATHS

```
Data: graph, start_node, ref_chain, iterations
Result: Paths - set of all paths, which start in start_node and end
the in ref_chain
Paths ← empty set
i ← 1
while i ≤ iterations do
  next_node ← select random node connected with start_node
  path ← [start_node]
  while next_node not in ref_chain do
    if next_node not in path do
      add next_node to path
    next_node ← select random node connected with the last node in the
path
    if all nodes connected with next_node are in path do
      path ← [ ]
      break
  if length(path) > 1 and path not in Paths do
    add path to Paths
  i ← i + 1
return Paths
```

**Algorithm 2**: COMPUTE COMPLEXITY

```
Data: graph, reference organism, window, number of iterations
Result: complexity values for each node in the reference
ref_chain ← reference nodes chain
set initial complexity values for all nodes as 0
for each node in ref_chain do
  Paths ← FIND PATHS(graph, node, ref chain, number of iterations)
  for each path in Paths do
    start ← first node in the path
    end ← last node in the path
    distance ← |position(start) - position(end)|
    if distance ≤ window do
      for each node between start and end do
        complexity[node] ← complexity[node] + 1/window
return complexity values
```

The algorithm has the following user-defined parameters: *window*, which is the size of the area around node X to which deviating paths should be connected (default 20 nodes), and *iterations*, which is the number of random walk processes from each node (default 500).

## Hotspot definition

Genome complexity profiles often contain peaks that are surrounded by regions with relatively low values. This complexity peaks correspond to the regions with high local variability, called variability hotspots. Genes are considered to belong to a hot-spot if their complexity exceeds a threshold, defined as the third quartile value plus the interquartile range, multiplied by the coefficient, $k$ ($k$ equals 1.5, by default), which is an arbitrary but commonly used criterion for outlier detection, initially proposed by Tukey [15]. The coefficient, $k$, can be altered by the user, to obtain the only highest values or to include modestly complex regions. Hotspot region coordinates can be downloaded in the web-based version and can be obtained in the command line version. Because no rigid mathematical definition for hotspots exists, users can

infer them with their preferred methods and thresholds, after downloading the complexity values from a GCB web site or calculating them using the stand-alone, command-line version.

## Simulations of genomes with predefined variability profiles

We hypothesized that the calculated complexity values observed for some regions of a chromosome correlated with the frequencies of fixed rearrangements in that region. To verify this assumption and to validate the algorithm, sets of model genomes were generated. These simulations included random gene insertions, deletions, HGT events, and inversions. HGT and random insertion probabilities were set as equal to the deletion event probability, to maintain the genome length. The probability of inversions was set to 1/100 the number of other events, based on data in the literature reporting that inversion events are less common than other types of rearrangements, such as deletions and duplication [16]. The localization of these changes throughout the chromosome was determined according to the predefined variability profiles. Next, these model genomes were processed by the complexity computing algorithm, and the results were compared with input distributions (more details are available in S1 Text). The algorithm for simulations can be found in the S2 Listing.

## Subgraph generation

To visualize a gene context in the region of interest, a subgraph representing this region can be constructed. First, a subset of reference chain nodes, representing the region of interest, is added to the graph. Next, the algorithm iterates through other genomes in the set and adds deviating paths that are limited to the selected region. If the length of the path is greater than the *depth* parameter, then the path is cropped, and only the start and end fragments (tails) of a fixed length (*tails* parameter, *tails* < *depth*) are added to the subgraph. If the weight of any edge is less than the user-defined *minimal_edge_weight* parameter, this edge is not added to the subgraph. To generate a subgraph, it is necessary to set the reference genome, the *start* and *end* coordinates (in base pairs) of the region. The subgraph generation algorithm can be found in the S3 Listing.

## Web server data acquisition and preparation

To construct a dataset for the webserver, we downloaded genomes for 143 prokaryotic species with more than 50 genomes available in the RefSeq database. For each species, if the number of complete available genomes was higher than 50, then only complete genomes were used. If the number of available genomes was higher than 100, then exactly 100 genomes were randomly selected for further analysis. The only exception was the *Escherichia coli* extended genome set, which contained 327 complete genomes, as of November 2017.

All downloaded genomes were reannotated with Prokka ver 1.11 [17] to achieve uniformity. Genes were assigned to orthogroups with OrthoFinder ver. 2.2.6 [18]. Python scripts contained within the GCB application were used to parse OrthoFinder outputs, calculate genome complexity values, and generate subgraphs around genome regions of interest.

## Additional methods

To obtain a phylogenetic tree of different species of the Bacillus genus, we aligned protein sequences with muscle [19], converted them into codon alignments with pal2nal [20], and built tree by iqtree with ModelFinder Plus option; snakemake pipeline for these steps is available at https://github.com/paraslonic/orthosnake/blob/tree/Snakefile_tree. All other phylogenetic trees were inferred with Parsnp v1.2 [21]. Retention indexes were calculated using the RI

function from R phangorn library [22]. To estimate the similarity to the reference genome, as in the analysis presented in Fig 6, all genomes were aligned with nucmer [23], and a similarity score was calculated as follows: all aligned reference genome ranges were reduced with IRanges R package [24], their total lengths were divided by the reference genome length, and all query genomes were sorted by this value, after which the strains with the highest values were chosen. Nucmer was used to detect synteny blocks between genomes from the same species, and Mauve [25] was used to detect synteny blocks between genomes from different species. Prophages were detected with Phaster [26]. To obtain Fig 3 we exported graphs in JSON format from GCB and visualized them in Cytoscape [27]. For Fig 3A, we used GCB, with the following parameters: *tails* = 1 and *minimal_edge_width* = 5. To produce Fig 3B, we used GCB, with the following parameters: *tails* = 1, *minimal_edge_width* = 5. To produce Fig 3C, we used GCB, with the following parameters: *tails* = 0 and *minimal_edge_width* = 5. The code used to generate Figs 4-7 is available at https://github.com/paraslonic/GCBPaperCode.

## Results

### Software description and availability

We have developed a method that represents the neighborhood of genes in a graph form and which estimates the local variability of the genome based on the resulting graph (Fig 1). We implemented this method in the GCB tool, which is available as both a standalone application and a web server. The GCB web server is located at https://gcb.rcpcm.org and contains data for 143 prokaryote species. A subset of the available genomes was included in analyses for which the number of available genomes was greater than 100 (except for *E. coli*). The complexity profiles in the web version were calculated with window sizes of 20, 50, and 100 genes. If a user desires to perform an analysis on a custom set of genomes or with different window sizes, then the standalone version should be used. All features of the web version are also available in the standalone version. To use the standalone version, a user should have basic command-line program operating skills. No precalculated data is available in the standalone version.

 **Elements of graphic user interface.** GCB browser-based GUI consists of three primary components: 1) the complexity plot; 2) the subgraph visualization; 3) the left sliding panel (Fig 2A).

 In the left sliding panel, the user can select the genome set (one per species in the web server, an arbitrary set in the standalone version), a particular genome, and a contig (in the case of a draft genome or when a genome consists of several replicons). When a genome has been selected, the complexity profile of the selected genome will be plotted in the *Complexity plot* panel. In the left sliding panel, the user can also specify the coordinates for a specific region of interest, which will be visualized in a graph form. The size of the region should not exceed 100 kilobases, to enable the performance of the graph visualization step. The user can search gene annotations to identify the locations of genes of interest, using the *Search* tab in the left sliding panel. Searching is performed over the *product* features of annotated genomes (only protein-coding sequences are considered).

 The *Complexity plot* panel shows a visualization of the complexity profile for the selected genome. In the *File* tab of the left sliding panel, numeric values associated with the complexity profile can be downloaded as a text file, for further analysis (e.g. comparisons against other profiles for other organisms). The visualization of custom data can also be added in the *File* tab of the left sliding panel. This may be useful to visualize GC content, pathogenicity islands, prophage regions, and sequence motifs along with a complexity profile. Features should be in a file, one line per feature, formatted as: <position><TAB><numeric value>. To draw a subgraph around the genome region of interest, the user can click in the middle of this region and

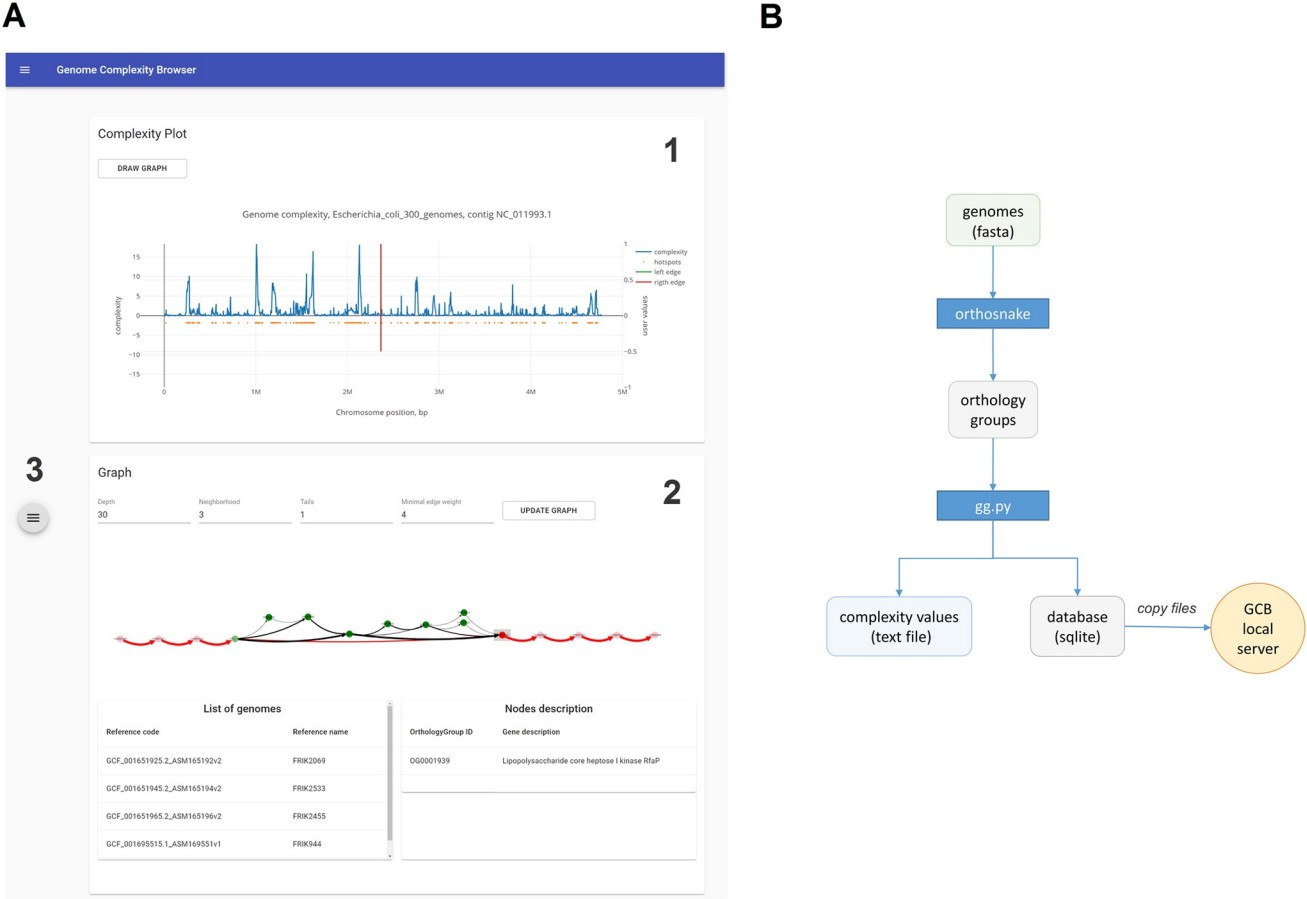

**Fig 2. Screenshot of the GCB browser-based interface.** The GCB GUI is available at gcb.rcpcm.org and when GCB is run as a local server. The GUI consists of a complexity profile panel, subgraph visualization panel, with a box containing information regarding nodes and edges, and a left sliding panel, in which the user can select an organism, set parameters, save and load files.

push the *Draw graph* button. The graph-based representation of the selected genome region will appear in the graph panel. To adjust the width of the region of interest, the user must change the value of the *Neighborhood* setting, located on the top in the graph panel, and also available in the *Other settings* tab of the left side panel. Genes for which the complexity level exceeds the threshold (belonging to hotspots) are marked with orange dots below the complexity profile. This marks can be disabled in the *Other settings*. Hotspots are also indicated in a text file with complexity values.

The *Graph* panel shows a graph-based representation of a selected region of the genome. Several settings are available for customizing the subgraph visualization, to simplify the analysis. Increasing the *Minimal edge weight* settings allows hiding the edges that are present in a small number of genomes (especially useful when rendering hot spots). Long paths in the graph complicate the analysis, they can be cropped to fragments of the *Tails* length; the maximum length of non-cropped paths is set by the *Depth* parameter. Subgraphs can be saved as JPEG images or exported in JSON format, which can be visualized using specialized software (e.g., Cytoscape) for the preparation of publication-ready images. This can be done in the *File/Graph* tab of the left sliding panel.

**Standalone version.** For the analysis of a custom set of genomes, we have provided pipelines that will enable users to perform the following tasks: 1) infer orthogroups, 2) generate a

 

graph, 3) estimate complexity values in text form and an SQLite database file, 4) generate and visualize subgraph. Database files can be imported to a local GCB server, which can be run on a standard PC. Fig 2B shows a roadmap for the standalone analysis.

Fasta-formatted genomes are expected as the inputs (both complete and draft assemblies may be used, although, we recommend the inclusion of at least one complete genome to be used as a reference). Snakemake pipeline (https://github.com/paraslonic/orthosnake) is provided to infer orthogroups. It performs genome annotation with Prokka [17], generates protein sequences in Fasta format, with position and product information in the header, and makes orthogroup inferences with OrthoFinder [18].

Python scripts, which are available at https://github.com/DNKonanov/geneGraph, should be used for further analysis steps. The OrthoFinder output serves as the input for the gg.py Python 3 script, which produces an SQLite database, a sif-formatted file containing the graph-based representation of all of the genomes, and complexity profile for the specified reference genome (for each of the genomes if no reference was set). Optionally, the analysis can be limited to a subset of genomes by providing a text file containing genome names, which can be useful for analyzing the variability of a specific subset of genomes (e.g., phylogenetic clade). Documentation and tutorials are available at https://gcb.readthedocs.io.

To estimate the genomic variability profile, the number of genomes should not be too small, a few dozens or hundreds are typical values. The upper limit depends on the computational resources available to infer orthogroups, which is the most computationally difficult step.

## Subgraph visualization

Graph-based representations of gene order can provide convenient methods for visually inspecting the contexts of genes of interest and to identify conservative and variable gene combinations. GCB can construct and visualize subgraphs, which are those parts of the genome graph that contain the region of interest. Next, we will describe examples of subgraphs generated using GCB and visualized with Cytoscape [27].

**Subgraph visualization reveals conservative and variable parts of operons.** Fig 3A shows a subgraph representing the gene context of the capsule gene cluster (chromosomal coordinates 3111444-3128026 in NCBI sequence NC 011993.1) for 327 complete *Escherichia coli* genomes. From this visualization, the operon can be observed to contain two conserved regions and one variable region. The variable region consists of genes from the serotype-specific synthesis region, whereas the neighboring conserved regions correspond with regions involved in polysaccharide export [28]. The capsule is considered to be an important virulence factor [29] for *E. coli* and many other bacterial species, and capsule variations are essential for the avoidance of immune responses and phage infections [30, 31].

Although the existence of variable and conserved regions of this operon was previously known, new information regarding the architecture of other operons may be obtained through analyses performed in GCB.

**Subgraph visualization reveals a genome variability in a particular locus.** Fig 3B and 3C show the visualization of subgraphs for regions containing two operons: hemin uptake (hmu) and propanediol utilization (pdu). The presence of E. coli harboring these operons in the intestinal microbiome has previously been associated with Crohn's disease [32–34]. These operons have different phylogenetic distributions, with the hmu operon being preferentially present in the B2 phylogroup (S2A Fig, retention index = 1), and the pdu operon can being found in phylogenetically distinct strains of *E. coli*, where its presence is in low agreement with the phylogenetic tree (S2B Fig, retention index = 0.26). The hmu operon is located at the

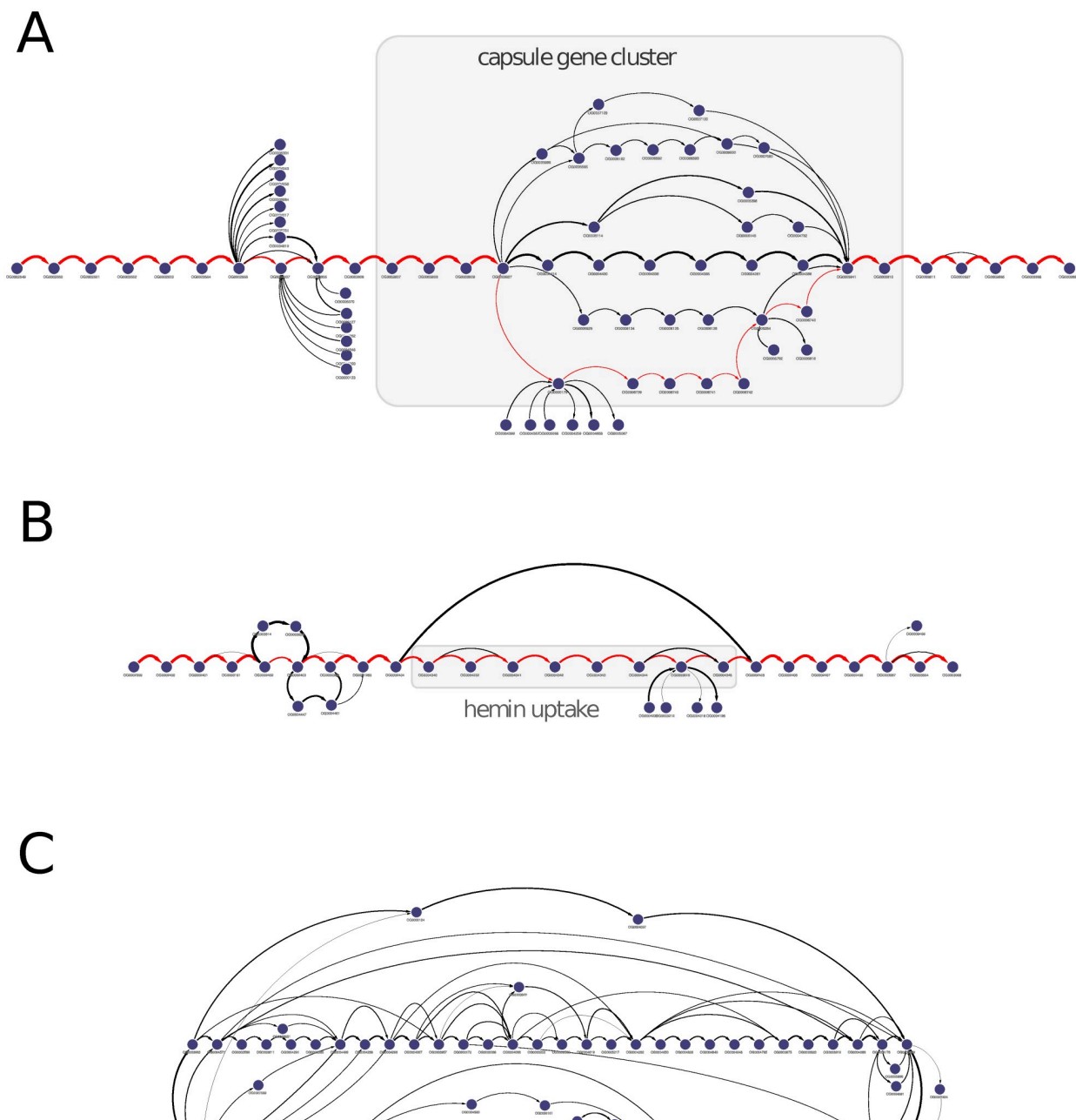

**Fig 3. Subgraph visualization allows the identification of conservative and variable operon regions and can reveal genome variability in a particular locus.** Subgraphs for the regions containing: A) the capsule gene cluster, B) hemin uptake, and C) propanediol utilization operon. A) Graph-based visualization reveals the conserved and variable regions of the capsule operon. Visualizations in B) and C) indicate that both the hemin uptake and propanediol utilization operons are located in conserved regions. Genomes that do not contain hemin uptake operons do not contain other genes in the same region. In several genomes, where the propanediol utilization operon is located, alternative and highly variable gene sets can be found.

3691615-3700567 positions and the pdu operon is located at the 2083448-2101340 positions of the NCBI Reference Sequence NC_011993.1.

The graph-based visualization reveals that the hmu operon is located in a conserved region, in which the neighboring genes are the same in all strains in which this operon can be found (Fig 3B). The edge that bypasses the operon indicates that in some genomes, the genes to the left and right of the operon are adjacent. Graph-based visualization also indicates that one of the genes (hemin transport system permease, HmuU) or close homologs are present in two alternative contexts.

The pdu operon is harbored by only a fraction of all considered strains (27 out of 327) but is also located in a conserved region (Fig 3C). Some variations in the pdu operon are visible and reflect different operon variants [34]. Unlike the hmu locus, here alternative gene sets are present. These alternative sets include genes associated with iron transport (FepC, FcuA, and HmuU), DNA mobilization (retroviral integrase core domain and the transposase DDE Tnp ISL3). These alternative sets can have high variability, with many overlapping changes observed in the subgraph visualization.

In the next section, we will describe the quantitative measurement of this observable difference in subgraph complexity.

## Complexity is a measure of genome variability

In a set of genomes with identical gene contents and localizations, each node in the resulting graph will have two edges. Any gene rearrangements (deletion, translocation, and insertion) result in the addition of new edges. We hypothesized that the number of distinct paths in a subgraph that represents a genomic region will monotonically depend on the frequency of the fixed gene rearrangements in that region.

We implemented an algorithm (Algorithm 2, described in the Materials and Methods) to count the number of distinct random walks in a subgraph that represents a given region of the reference genome, and the computed value is referred to as region complexity. By selecting subregions with a sliding window and calculating the complexity value for these subregions, we can obtain the complexity profile of the reference genome. The size of the sliding window can be set by the user, and the default value of 20 was used for the results described below.

**Method verification.** We verified our approach by performing genome evolution simulations based on potential gene transfer events, by comparing with previously published results and through the analysis of known variability hotspot regions (integrons).

The complexity profile for *E. coli*, as calculated by GCB, is in good agreement with the hotspots evaluated in [35] (Fig 4A). Complexity values within the hotspots identified in [35] were significantly greater than those outside of the hotspots (p-value $< 10^{-16}$, Mann-Whitney U test, see Fig 4B).

We performed several simulations, during which we suggested that the probability of genomic rearrangement events (HGTs, deletions, and translocations) was non-uniformly distributed along the chromosome, which may reflect the unequal probability of changes or their fixations. The algorithm for performing simulations is listed in the S2 Listing. We used three patterns to generate profiles of such probabilities, including sinusoidal, rectangular, and sawtooth, and performed 10 independent simulations for each pattern. The number of rearrangement iterations was 3,000 for each model. The results of our method were in good correspondence with the predefined distribution (R-squared $> 0.74$, Spearman correlation $> 0.69$, FDR corrected p-value $< 10^{-300}$, Fig 3C). Results for each simulation are available in the S1 Text.

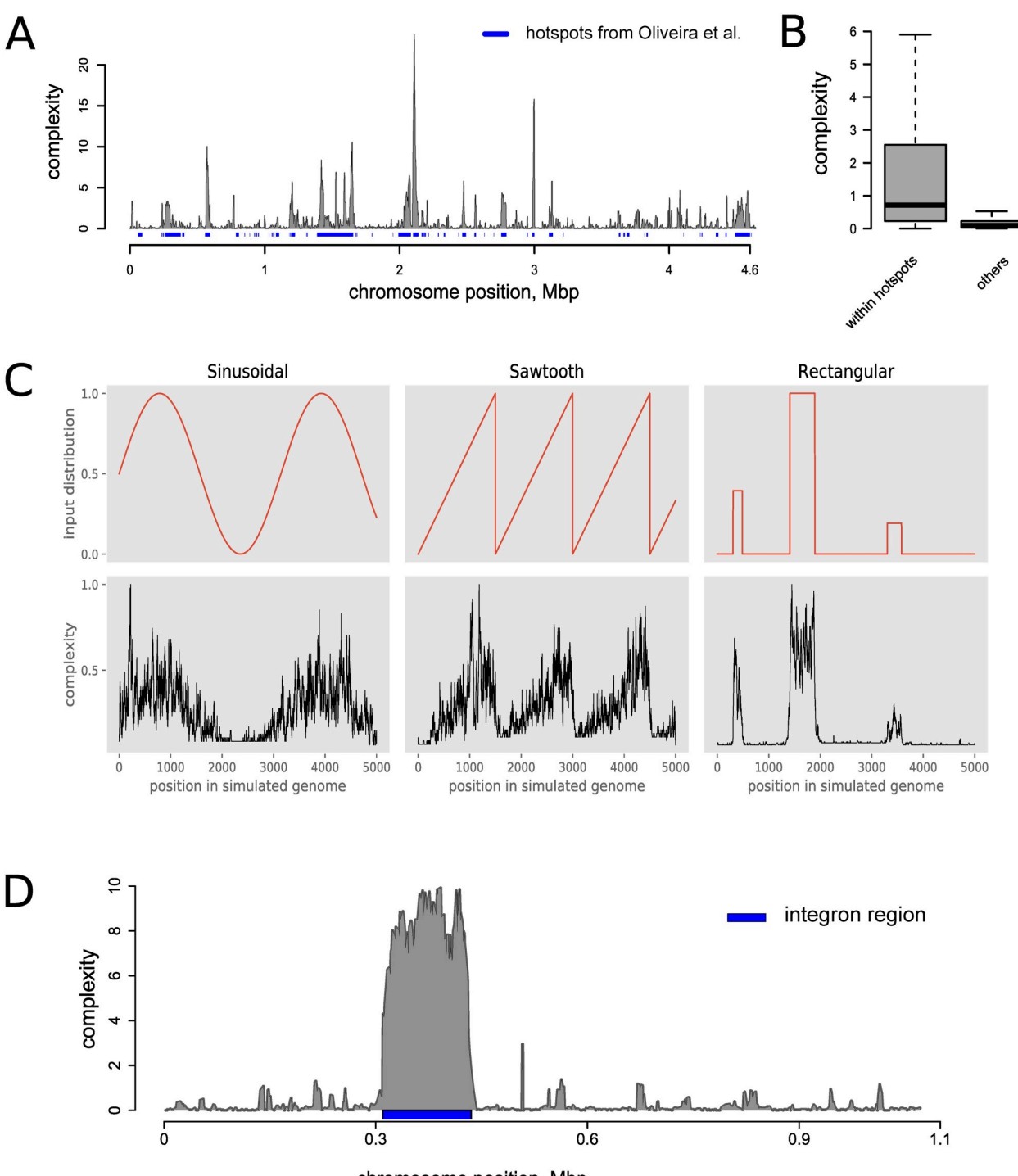

**Fig 4. Complexity values can be used as genome variability measures.** A) Comparisons of complexity profiles for *E. coli K12*, with hotspots identified in [35] (blue rectangles underneath complexity profiles). B) Comparisons of complexity values among genes located inside and outside of the hotspots identified in [35]. C) Comparisons between the initial variability profiles and the complexity profiles that were calculated based on the artificial genomes, after 3,000 evolution simulation steps. D) Complexity profiles for *Vibrio cholerae N16961*, chromosome II, with noticeably high levels of complexity at the integron regions.

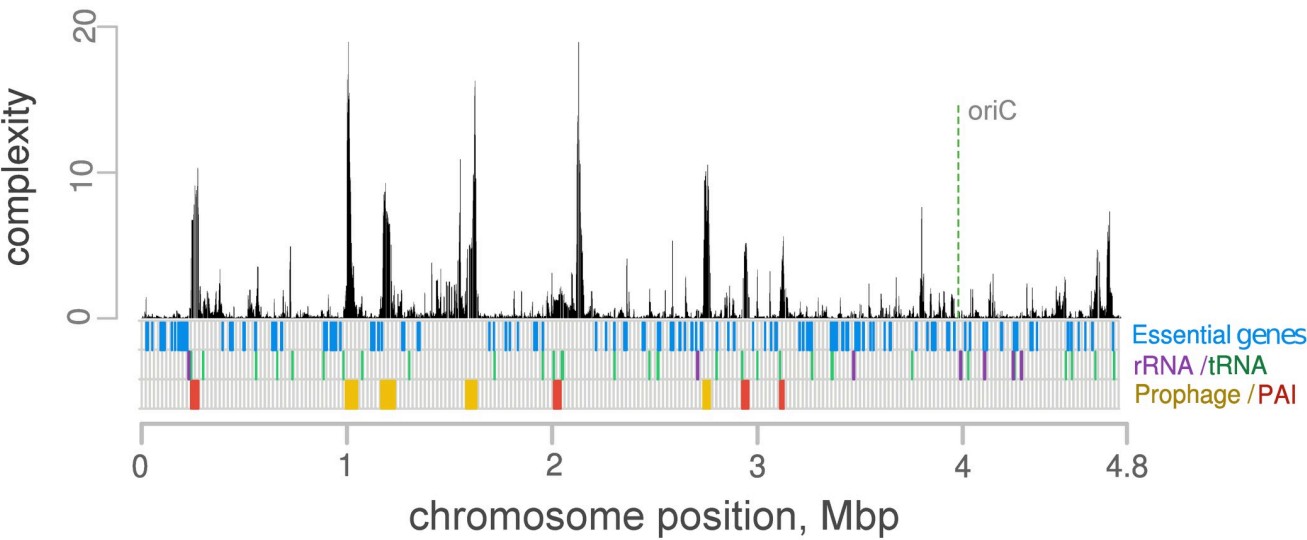

**Fig 5. Regions with high complexity values are mostly associated with mobile elements.** The complexity profile for the *E. coli LF82* chromosome is shown, and 327 other *E. coli* genomes were used to calculate complexity values. Orange-colored bars denote prophages and red-colored bars denote pathogenicity islands. For essential genes, low complexity values are observed. Some of the most variable regions lack the features of mobile elements.

Integrons are gene acquisition systems that are capable of the integration, excision, and rearrangement of gene cassettes and are examples of genome variability hotspots. We observed that integron regions have high complexity values. Fig 3D shows integron region of *V. cholerae* as an example. This integron was dubbed a superintegron because of its high length of about 120 kbp and more than one hundred integrated cassettes [36].

**High complexity values are associated with prophages and genomic islands but not limited to them.**    Fig 5 shows the complexity profiles along the chromosome of adherent-invasive *E. coli LF82* [37]. As expected, the regions with a high density of essential genes have low complexity values. In contrast, pathogenicity islands and prophage regions have relatively high complexity values. Some chromosomal loci with high complexity values have no recognizable signs of mobile elements. A subgraph of the region with the highest complexity values (located at 2,115,791-2,164,382) does not contain recognizable genes with mobility associated functions.

**Complexity profiles have conservative features at both the inter- and intraspecies levels.**   Genome complexity analyses can be used to compare variability profiles among different species or intraspecies structures (e.g. phylogroups). We performed comparisons of the complexity profiles for 146 species and observed that when genomes are sufficiently similar (synteny blocks cover most regions of the genomes), complexity profiles are also similar (S3 Fig).

Fig 6A shows comparisons among the complexity profiles of four *Bacillus* species, and Fig 6B shows their phylogenetic relationships. Regions with high complexity values are associated with prophages (denoted by the orange bar below the complexity profile) and have conservative locations. Some regions that lack integrated viruses also have high complexity values and are conservatively located in the genomes of different species (i.e., the region located at 2.5 Mbp in *B. subtilis*), whereas others are only highly variable in one species (e.g. the region located at 2.8 Mbp in *B. velezensis*).

Fig 6C shows a comparison of the complexity profiles for different E. coli phylogroups [38]. For each of the five large phylogroups (A, B1, B2, D, and E), we selected one reference strain and the 100 most similar strains among 5,466 RefSeq genomes (both finished and draft assemblies). Fig 6D shows a phylogenetic tree for the selected genomes. Complexity profiles for each

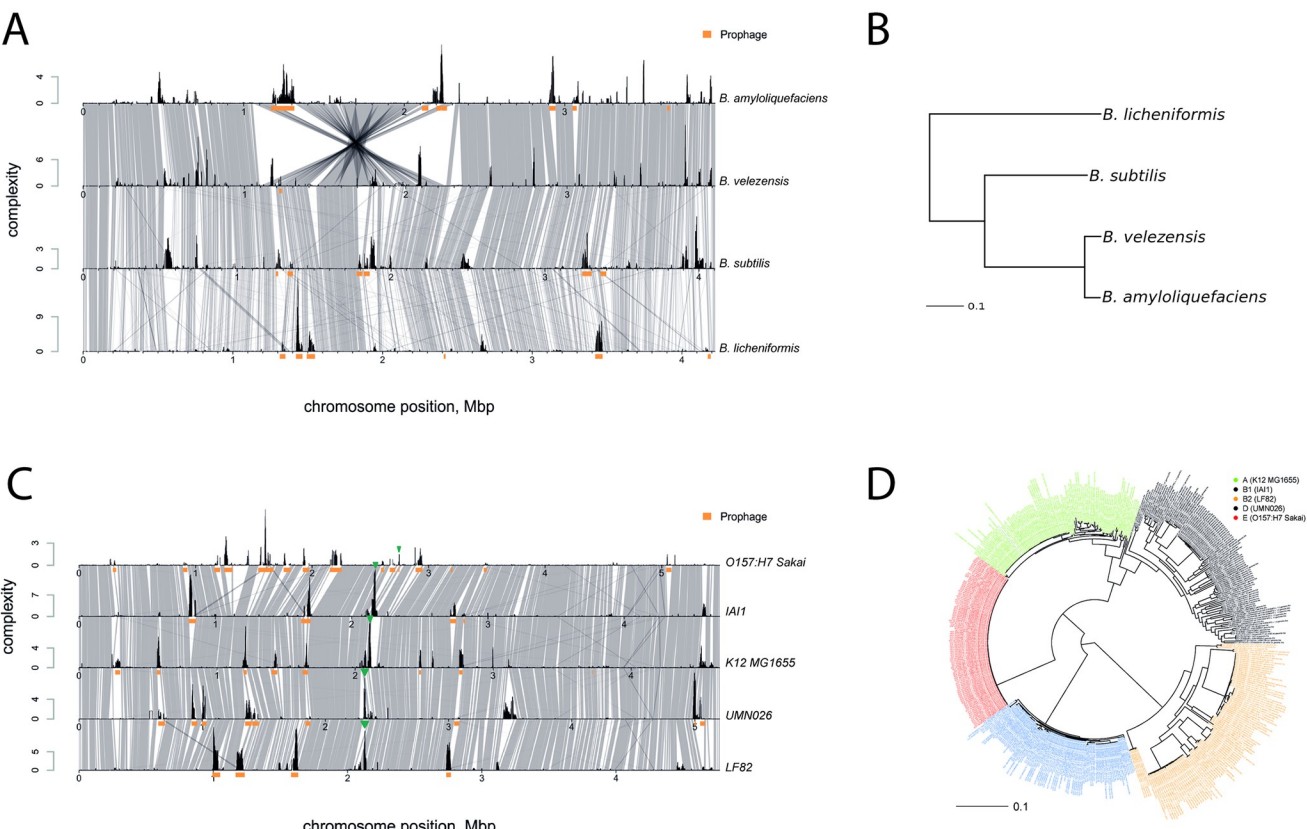

**Fig 6. Regions with high complexity values are primarily located in conserved regions, when both intra- and interspecies comparisons are performed.**
A) Complexity profiles and synteny blocks for the four *Bacillus* species. B) Phylogenetic tree for the four *Bacillus* species. C) Complexity profiles and synteny blocks of the five *E. coli* phylogroups, in which green triangles denote conserved regions with high variability that are not associated with prophages and genome islands. D) Phylogenetic tree of the *E. coli* genomes selected for this analysis. For each phylogroup, one reference strain and the 100 closest genomes were selected.

reference genome were inferred, using genomes from the corresponding clade, only. This comparison revealed that many of the regions with high variability rates similarly located in genomes of strains belonging to different phylogroups. The majority of these regions contain prophages, but some do not include phage-associated genes. Transient hotspots (with high complexity values in some clades and low complexity values in others) can also be observed.

We described a region in the *E. coli* genome with a high variability rate, without identifiable mobile genetic elements (designated with a green triangle in Fig 6C). Fig 6C shows that this variability hotspot is present in the A, B1, B2, D, and to a lesser extent, E, phylogroups. Phylogroup E consisted of genomes that were closely related to the *O157:H7 Sakai* strain and contained the largest genomes, primarily due to the expansion of bacteriophages. We observed that only in this phylogroup did this high-variability region contain integrated prophages. Prophage integration can explain the variability in the E phylogroup; however, the driving force behind high variability in this region for the other phylogroups remains to be elucidated.

## Method applicability

Complexity profiles and subgraphs could be obtained for any set of genomes for which orthogroups could be inferred. The graph-based representations, window-based variability estimation, and subgraph visualizations results are optimized when the set contains closely

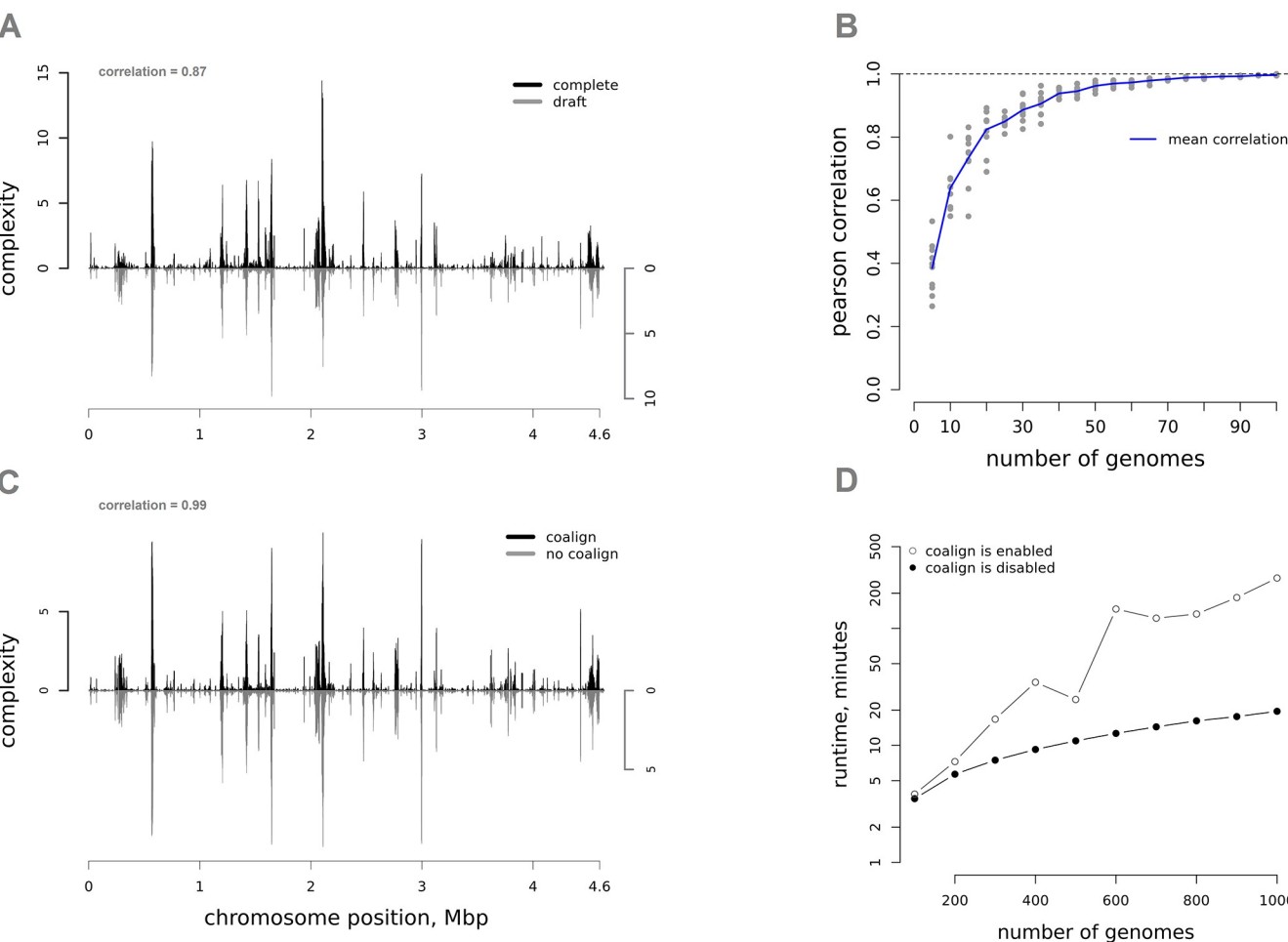

**Fig 7. Method applicability and benchmark.** A) Complexity profiles computed with 100 complete genomes (at the top) or 100 draft genomes (at the bottom) for *E. coli* are very similar, B) Correlations between the complexity values obtained with either 100 *E. coli* genomes or for subsets with lower number of genomes. C) Complexity profiles computed with and without *coalign* option for 100 *E. coli* draft genomes are almost the same. D) The time needed for graph construction and complexity estimations, based on different numbers of genomes, with or without *coalign* option.

related genomes, in which local variability is not overwhelmed by chromosomal rearrangements longer than the chosen window size. From our experience, and the estimates from other studies [39], genomes within 0.05 phylogenetic distance (approximately, a species boundary [40]) should be used. The complexity profiles of different sets of genomes can be compared with the same limitations, and large amounts of genome rearrangements can result in uninformative comparisons. We observed that some species have display no obvious peaks in their complexity profiles (for example, the naturally competent *Pseudomonas fluorescens*), which makes comparative analyses less informative than comparisons for species with clear regions of low and high complexity, see S4 Fig. Meanwhile naturally competent *Pseudomonas aeruginosa* provides quite clear and comparable profiles (S4 Fig).

Draft genomes (consisting of fragmented genome regions, called contigs) may be used for complexity estimations. We performed comparisons of complexity profiles, inferred from either 100 complete or 100 draft genomes, using the same complete genome as a reference, and observed significant similarity (Pearson's correlation coefficient is 0.87, Fig 7A). The impacts of draft genomes may be higher for highly stable genomes for which large-scale

rearrangements represent the primary source of variability. Subgraph visualizations can suffer from genome fragmentation because false negatives may be introduced by contig boundaries (for example, the context of a region representing some particular contig may not be able to be identified).

Small numbers of genomes included in the analysis may lower the accuracy of complexity profile estimation. We recommend the use of no fewer than several dozens. Fig 7B shows the correlation of complexity values that were obtained with either 100 *E. coli* genomes or subsets containing lower numbers of genomes. When more than 40 genomes are included in the analysis, Pearson's correlation coefficient becomes greater than 0.9.

The time required for the analysis depends on the number of genomes being analyzed. Fig 7D shows the time required for the graph construction and complexity evaluation steps for different numbers of draft genomes (up to 1,000). The analysis was performed on laptop with i7 9750h CPU. The maximum memory usage was 2.3 gigabytes. The runtime differed significantly with coalign option turned on (in this case, the contig orientation is selected to best match each other), or turned off. When coalign was enabled, analysis time grew quadratically with the number of contigs used in the analysis. Meanwhile, the complexity profile, with this option enabled, is almost identical to the complexity profile, computed with disabled coalign (Pearson's correlation coefficient greater than 0.99). The analysis time for full genomes will not vary significantly when coalign option is turned on or off. The most time-consuming step for the overall analysis is the orthogroup inference. Inference of orthogroups for 1,000 *E. coli* genomes took five days on our computing cluster (50 threads on AMD EPYC 7502 2x32).

Local variability hotspots that are identified by GCB often coincide with prophages and genomic islands. We compared the identified hotspots with a curated, literature-based dataset [41] and obtained a mean F1 score of 0.65, which is comparable to existing tools, with a mean precision score of 0.54. Comparisons with the automatically generated dataset results in much lower accuracy scores. Detailed information can be found in S2 Text. We concluded that complexity analysis can assist in the explorative analysis of genomic islands or prophages, but should not be used for genomic island predictions without additional analysis, because hotspots may be due to different (often unknown) origins.

## Discussion

Synteny visualization tools (e.g. Mauve [25], BRIG [42], and genePlotR) are often used for genome comparative studies and allow the visual inspection of large and small genome rearrangements. The number of genomes that can be effectively compared using existing approaches ranges from a few to several dozens. However, hundreds and even thousands of genomes are currently available for some species, and these large amounts of genomes can be used to gain new information regarding genome variability, genome architecture, and operon structures. To efficiently analyze large sets of genomes, we proposed a graph-based approach, in which genes are represented using nodes that are connected depending on their co-localization (neighborhood).

Graphs were previously applied to the analysis of genome changes, in the form of breakpoint graphs [43], which can be used to reconstruct possible ancestral states but are not convenient for visualization properties in our opinion. They have also been used to represent known genome variants, to increase mapping quality [44]. To our knowledge, gene neighborhood graph visualization is only available in the FindMyFriends R, other than in GCB.

Graph-based representations of a set of genomes and the selection of subgraphs that represent regions of interest can facilitate the answering of the following questions. Is a gene

(operon) located in the same location in all genomes? If not, then what alternative genes are present? Which parts of a gene set (operon) are conserved and which are variable? Which genomes contain some particular combination of genes?

Genome variability hotspots have been described for several bacterial species. In [35], the authors analyzed HGT hotspots for 80 bacterial species. They concluded that many hotspots lack mobile genetic elements and proposed that homologous recombination is primarily responsible for the variability of those loci. The factors that determine the locations of hotspots, their emergence, and elimination, remain open questions.

We implemented a method for the quantification of local genome variability, based on the number of unique paths in a subgraph. To our knowledge, GCB is the first tool that allows the quantification of genome variability based on a user-defined set of genomes. GCB provides a method for studying the dynamics of variability associated with hotspots, including changes in intensity and location on different levels, ranging from intraspecies structures, such as phylogroups or ecotypes, to interspecies and intergenus comparisons.

We compared variability profiles between different species and, in the case of *E. coli*, between different phylogroups. We observed that, as a rule, when genomes are close enough for the large synteny blocks to be detected (with blast or the nucmer tool), then complexity profiles appear similar, and regions with high complexity values are surrounded with low complexity regions, forming the similarly conserved regions in different groups of organisms. The analysis of the complexity profiles of *E. coli* revealed that many hotspots are located in prophage or pathogenicity island integration sites, and site-specific mechanisms could govern their conservative locations. Some hotspots lack such factors, and the reasons for their conservative locations remain to be elucidated.

The approach proposed here is not universal. For example, this approach is not suited for the detection of large genomic rearrangements (larger than the window parameter, which is usually several dozens genes) or changes in noncoding parts of the genome. Our methodology has also some limitations due to the dependence on orthology inference accuracy. Here, we used the OrthoFinder tool [18], which uses an MCL graph clustering algorithm, based on gene-length-normalized blast scores. We find this tool to be optimal in terms of efficiency and accuracy. However, paralogous genes may be attributed to one orthogroup, which can make graph-based representations of the context problematic. We observed that, on average, 0.5%, of all orthogroups per genome contain at least one paralogous gene; however, among all orthogroups inferred for the species, the proportion of orthogroups with paralogs is almost 16% (see S1 Table for information for each species individually. The code is available at https://github.com/paraslonic/GCBPaperCode/tree/master/GeneCount). We implement two possible methods for addressing paralogous genes in GCB: the default approach is to ignore them, whereas the other approach is to perform an artificial orthologization process (each paralogous gene with a unique left and right context is denoted with a suffix and added to the graph). From our experience, the optimal strategy is to utilize the default mode for explorative analysis, followed by the verification of all conclusions in the orthologization mode. The graph layout process is also difficult to automate. We used two layout algorithms (Dagre and Graphviz), but manual manipulations were often necessary to ensure a clear layout, and Cytoscape (or other graph manipulation software) is desirable for formatting publication-ready images.

Despite the above-mentioned drawbacks, we find that this proposed method of complexity analysis can be informative, as it successfully identifies known rearrangement hotspots (such as prophages and integrons), and we hope that GCB, with its capacity for both visualization and complexity assessment, will be applied to the area of comparative genomics studies.

## Conclusion

We have developed a novel tool, called the Genome Complexity Browser (GCB), that can analyze sets of genomes for the quantification of genome variability and to visualize gene context variants. We used graph-based representations of gene neighborhoods to visualize and estimate local genome variability. The GCB browser-based interface enables the simultaneous analysis of genome variability profiles and patterns of changes that occur in a particular locus.

We precalculated data for 143 prokaryotic organisms, and the webserver gcb.rcpcm.org can be used to browse these genomes. A command-line tool and a stand-alone server application allow the user to analyze any particular set of genomes.

We observed that genome regions with high variability exist, which are surrounded by conserved regions when examined by both intra- and interspecies comparisons. Some of these regions lack any genes with identifiable mobility functions.

## Supporting information

**S1 Fig. Comparison of the two methods for addressing paralogs.** A) The graph obtained with the default approach, which ignores groups with several representatives in the particular genome. B) The graph obtained using the paralog "orthologization" approach.
(PDF)

**S2 Fig. Phylogenetic tree of 327 E. coli strains, with information regarding the presence of pdu and hmu operons.** Red bars denote genomes in which the complete gene of the operons is present, green bars denote genomes in which more than half of the operon genes are present. A) The hmu operon is in good correspondence with the phylogenetic tree of *E. coli*. B) The pdu operon presence is poorly correlated with the phylogenetic tree of *E. coli*.
(PDF)

**S3 Fig. Comparisons of the complexity profiles of 146 prokaryotic species.** Complexity profiles are shown on the same scale for all organisms. Synteny blocks are shown in green. The phylogenetic tree was built based on the 16S rRNA sequence.
(PDF)

**S4 Fig. Comparison of the complexity profiles of phylogenetic clades for *P. fluorescens* and *P. aeruginosa*.** Complexity profiles and synteny blocks are shown on the left, phylogenetic trees are shown on the right.
(TIF)

**S1 Text. Simulations of genomes with predefined variability profiles.** Details concerning the methods and results of genome simulations, with predefined variability profiles, and further complexity analysis.
(PDF)

**S2 Text. Comparison of hotspots identified by GCB with a curated, literature-based and automatically generated datasets.** MCC, F1 score and precision values are given.
(PDF)

**S1 Listing. Algorithm used to coalign genomes in the set.**
(PDF)

**S2 Listing. Algorithm used to simulate genomes with predefined variability distributions.**
(PDF)

**S3 Listing. Algorithm used to generate subgraphs representing genome regions of interest.** (PDF)

**S1 Table. The proportion of orthogroups containing paralogs in 143 species.** Average values are given for a single genome within a species, and for the species as a whole. (XLS)

## Acknowledgments

We thank the Center for Precision Genome Editing and Genetic Technologies for Biomedicine, Federal Research and Clinical Center of Physical-Chemical Medicine of Federal Medical Biological Agency for providing computational resources for this project.

## Author Contributions

**Conceptualization:** Alexander Manolov, Dmitry Fedorov, Elena Ilina.

**Data curation:** Ivan Osmolovsky.

**Investigation:** Alexander Manolov.

**Methodology:** Dmitry Konanov.

**Software:** Alexander Manolov, Dmitry Konanov, Ivan Osmolovsky, Rinat Vereshchagin.

**Supervision:** Elena Ilina.

**Visualization:** Alexander Manolov, Dmitry Fedorov, Ivan Osmolovsky.

**Writing – original draft:** Alexander Manolov, Dmitry Konanov, Dmitry Fedorov, Elena Ilina.

**Writing – review & editing:** Rinat Vereshchagin.

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
