## [Decision Letter · Decision Letter 0]

24 Mar 2020

Dear Mr. Manolov,

Thank you very much for submitting your manuscript "Genome Complexity Browser: visualization and quantification of genome variability" for consideration at PLOS Computational Biology.

As with all papers reviewed by the journal, your manuscript was reviewed by members of the editorial board and by several independent reviewers. In light of the reviews (below this email), we would like to invite the resubmission of a significantly-revised version that takes into account the reviewers' comments.

We cannot make any decision about publication until we have seen the revised manuscript and your response to the reviewers' comments. Your revised manuscript is also likely to be sent to reviewers for further evaluation.

Sincerely,

Morgan Langille, PhD

Associate Editor

PLOS Computational Biology

William Noble

Deputy Editor

PLOS Computational Biology

Reviewer's Responses to Questions

**Comments to the Authors:**

Reviewer #1: In this study Manolov et al propose the ‘Genome Complexity Browser’ (GCB), a novel tool allowing visualization of gene contexts in a graph form and quantification of genome variability.

The manuscript is clear, relatively well-written, and fills a gap in the sense that it provides a relatively accurate measure of gene variability at a local level (by computing the complexity value).

I have the following observations / criticisms, which I hope will help the authors improve their manuscript:

1) The authors mention multiple times ‘hotspots’ as chromosomal loci in which HGT events are preferentially concentrated. In what measure (particularly via the complexity value), does GCB helps disentangling between regions of the chromosome that are (or are not) hotspots? Although valid, the complexity value computed by the authors will only be useful for the wide community if it is able to provide some clue about what constitutes a hotspot. In other words, above which threshold values of complexity can we call a region a hotspot?

2) The complexity values computed by the authors, also do not provide per se any information about the types of rearrangements taking place. For this, the user must analyze the graph representation. While the latter is useful for a trained user, it won’t be for many others. In this sense, it would be useful to split the complexity value into terms corresponding to replacements, gains, losses, etc. This would give the user a more immediate sense (and some quantification as well) of the frequency of these processes. It would also provide an idea if the region is, for example, subjected to high gene turnover, or is undergoing differential gene loss.

3) I presume that all the orthologous computed by the authors are positional ones? How does GCB handle, for example, genomic inversions?

4) Discussion: “On the other hand, it doesn't take into account phylogenetic information and syntenic relationships between different genomes, and erroneous homology inference sometimes occurs.” How often did this happen? The authors need to show some numbers. How many times paralogous genes were attributed to one group?

5) “Quantitative estimation of genome variability along the chromosome” -> “Quantitative estimation of variability along the chromosome”

Reviewer #2: Summary

The authors present here a novel tool entitled the Genome Complexity Browser to infer and visualize hotspots of genome variability across many prokaryotic genomes. The subject is of large interest to a broad audience and the software surely fills a gap in available tools for genome comparison. There are wide possible applications for this type of analyses, that are briefly covered in the result section. However, the manuscript suffers from the quality of the reporting and many aspects need to be clarified.

Major comments

Introduction

The first paragraph of the introduction should be improved to better explain the concept of genome architecture and constraints ruling gene order and genome rearrangements. “Regularities” is likely not the appropriate word here.

Materials and methods

The section misses a general description of the algorithm, eventually with a schematic figure, to allow a better understanding of the standalone tool: input, steps computed by the tool, output, visualization? Also, the authors must allow the reader to understand immediately the differences between the standalone tool and the webserver, including the type of available data.

The authors recommend that contigs of draft genomes are re-ordered to increase co-linearity before using their tools. What is the influence of having draft genomes on the accuracy of results obtained? It would be interesting to benchmark this aspect by comparing the results of a small set of complete vs draft genomes or reordered vs non-reordered genomes, highlighting and discussing potential biases. 

Results

Software description and availability:

The descriptions should be improved on all levels in the manuscript, with appropriate links provided, e.g. to the Snakemake script to infer homology groups (name, place?). The wording seems to imply that it can be used to identify homology groups, which could be understood as building groups of orthologs but does not seem to be the case. Please clarify.

“GCB browser-based GUI consists of three main parts”. Actually, there are four panels with the search panel. It should be described as well to respect the order in figure 2. Numbering the panels on the figure 2 would also improve reader's understanding at first sight.

Finally, a more detailed description of possible options in the manuscript (e.g. loading custom features to display) and on the website would be useful to the users (e.g. use and file format to input with “load file”).

The overall look of the webserver is disappointing, and common information for such tools are not available, such as FAQ, address/way to contact the developers or maintainers, etc.

The display of panel 1 is quite confusing. For example, if one fills OG, then coordinates must be updated (but the button appears a few lines below) and if one fills the coordinates, then OG must be updated (but the button is above). I would suggest revising the position of the different items so that users are directed on the correct button after filling the various fields.

The authors give many examples of how their tools could be useful to study variations in genome architecture within and among species. However, the description of examples and their corresponding figures is often very superficial, and leaves the reader with many questions. For example, how does the tool perform depending on the distance between genomes compared? As shown in some examples, it works well with highly collinear genomes. What is the effect of recombination prone bacterial species such as Pseudomonas on the assessment of genome variability? Can the tool be used efficiently to study more distantly related organisms, and under which conditions?

The authors elude to the possible use of their tool to identify prophages and genomic islands. Did they try to benchmark their tool with existing reference dataset such as that originally published by Langille et al (2008) and refined in Bertelli et al (2019)? This could be valuable to assess the usefulness of GCB compared to other existing tools.

Based on the github repository, the software seems to depend on an old version of OrthoFinder. Are the outputs of newer versions supported too? Also, the authors discuss that OrthoFinder does not take into account phylogenetic information, which contradicts the latest publication of OrthoFinder by Emms and Kelly (2019). Please be more accurate in the discussion.

Overall comments

The quality of written English must be improved throughout the manuscript as it contains many typo, grammar errors, incorrect words. For example, in “variable and conservative parts” the authors probably mean conserved, or “or it close homologs are present” (its), or simply poorly written sentences “Finally, integrons have expectedly high complexity values computed with here proposed method.”. The authors must make an extensive effort to improve the reporting style and the storytelling.

Minor comments

- “depending on the chromosome position” – I doubt the author mean the position of the chromosome, but rather the position of the gene on the chromosome.

- “ Local genome variability is estimated using a graph representation of gene order (neighborhood) with a here introduced measure called complexity.” Revise English

- Be careful of typing errors such as: “selection f a genome”

Reviewer #3: From what I can see Genome Complexity Browser is a visualization tool showing genome plasticity for complete genome (i.e. from refseq it seems). The concept is good, I think we need to move away from all-against-all comparisons like act, easyfig, mauve (because they wont cope with large datasets). And admittedly BRIG being reference-centric doesn't show the variability very well. AS far as I know there isn't a tool that does this out of the box (sure you can do a combination of BLAST, R, some networking library to get the same effect) so GCB addresses a clear need.

All in all my main criticism for this manuscript and the tool is that both do not help me, as a user, use it. There is a lot of information that is assumed. It is clear that the authors have not given this program to their colleagues to play around it. I suggest strongly that they do, under test conditions, where authors themselves do not offer any help. The tool should speak for itself and the it should be obvious to the user how to achieve certain tasks.

A lot more hand holding is required for this to be of general use. Firstly, with the interface. How am i to choose the OG and the coordinates? I assume the coordinates are in base pairs, but when i open an e. coli genome and enter start coord: 1 and end coord: 3000000. I presume the program will show that section of the genome, but it doesn't. It seems to snap back to some strange number (maybe this is dictated by the OG values in the field above). I don't really understand.

The Search bar, doesn't explain what I am searching. Locus tags? i dont know, there should be a suggested value/auto complete dropbox for the search box. What values would work, what values would not work. I don't know.

I played around with the program for quite some time, and I cannot figure out how to produce the figures they show in the manuscript.

It took me a while to figure out where complexity plot is (it's not shown by default and seems to be hidden).

What i am trying to point out, is that all the components are likely there, but I have no idea how to use it. Tool tips, more documentation, suggested values, worked examples, a demo, information in the manual are all absent.

For the website, The manual only explains the interface. It needs a step-by-step worked example to show people how to use it. Step-by-step, to the point where the manual says "Select X from the dropdown, then click button Y, ... " and so on.

It is likely that users will want to use genomes other than the few provided - but judging from the manual the process seems quite involved to do your own data. can the authors simplify this approach somehow?

In terms of the manuscript.

This tool is used to detect hypervariable regions which in turn implies mobile genetic elements. One example given here is prophage regions in E. coli and Bacillus. I can see the figures are annotated with prediction from PHASTER But it is not clearly stated what are the differences between this approach and what PHASTER predicts. Doing a comprehensive comparison would show what the benefits and limitations of GCB. I think the authors need to take more time with these worked examples to show the utility of their tool.

There is a question of scalabilty, how long would these graph traversals take if there was 1000 genomes for instance. The sample data seems quite small.

**Have all data underlying the figures and results presented in the manuscript been provided?**

Reviewer #1: Yes

Reviewer #2: Yes

Reviewer #3: Yes

PLOS authors have the option to publish the peer review history of their article (what does this mean?). If published, this will include your full peer review and any attached files.

Reviewer #1: No

Reviewer #2: No

Reviewer #3: Yes: Nabil-Fareed Alikhan
---

## [Decision Letter · Decision Letter 1]

5 Aug 2020

Dear Mr. Manolov,

We are pleased to inform you that your manuscript 'Genome Complexity Browser: Visualization and quantification of genome variability' has been provisionally accepted for publication in PLOS Computational Biology.

Best regards,

Morgan Langille, PhD

Associate Editor

PLOS Computational Biology

William Noble

Deputy Editor

PLOS Computational Biology

Reviewer's Responses to Questions

**Comments to the Authors:**

Reviewer #1: The authors addressed my concerns.

Reviewer #3: Thank you very much for taking on my suggestions. It is looking much better now and I could follow along with the tutorials and got some output that made sense to me.

Some extra suggestions as I went through GCB again.

* in terms of the complexity analysis, please add an extra output that will give a bedfile with the coordinates of highly complex regions (complexity > a value, maybe 2?). This will allow people mask MGE from SNP calling e.g. when they use snippy for core SNP phylogenies (https://github.com/tseemann/snippy) . Or they can use that to mark up genomic regions in Artemis for further inspection.

* Please add a SSL Certificate (see Let's encrypt for a free one) for the web service.

* the complexity plot on the website has a typo in the legend "rigth edge " > "right edge"

**Have all data underlying the figures and results presented in the manuscript been provided?**

Reviewer #1: Yes

Reviewer #3: Yes

PLOS authors have the option to publish the peer review history of their article (what does this mean?). If published, this will include your full peer review and any attached files.

Reviewer #1: No

Reviewer #3: **Yes: **Nabil-Fareed Alikhan

---

## [Editor Report · Acceptance letter]

2 Oct 2020

PCOMPBIOL-D-19-02244R1 

Genome Complexity Browser: Visualization and quantification of genome variability

Dear Dr Manolov,

I am pleased to inform you that your manuscript has been formally accepted for publication in PLOS Computational Biology. Your manuscript is now with our production department and you will be notified of the publication date in due course.

With kind regards,

Sarah Hammond
